# Channel Adjustments in Iranian Rivers: A Review

**Somaiyeh Khaleghi [1],\* and Nicola Surian [2]** 

[1]  Department of Physical Geography, School of Earth Sciences, Shahid Beheshti University,
     Tehran 1983969411, Iran
[2]  Department of Geosciences, University of Padova, 35131 Padova, Italy; nicola.surian@unipd.it
\*   Correspondence: s_khaleghi@sbu.ac.ir; Tel.: +98-2990-5620

**Abstract:** Channel adjustments in Iranian rivers have been intense over the last decades due to natural and human factors. Iran has six major basins, all with different climates, from very humid to very arid. This work is a review of the available studies and data about channel adjustments in Iranian rivers, and aims to reconstruct a first outline, at a national scale, of types, magnitude, and causes of adjustments. The results show that most of the rivers have undergone incision (1 to 2 m and, in some cases, up to 6 to 7 m) and narrowing (from 19% to 73%), although widening (from 22% to 349%) has occurred in some rivers. Narrowing is due to dams and sediment mining; widening is due to climate change and sediment mining. Incision is due to gravel and sand mining, dams, channelization, with in-channel mining being the main cause of incision. Channel adjustments have occurred in basins with different climates, but it seems that widening has been more intense in arid and semi-arid climates. Such adjustments have several negative effects (e.g., damage to bridges, degradation of river ecosystems, and instability of banks). The comparison between Iran and other countries shows that narrowing and incision have been the dominant processes in most of the rivers, while damming and in-channel mining have been used as the main controlling factors. Data about adjustments in Iranian rivers are neither homogeneous nor complete for all the rivers. This lack of completeness implies that our understanding of channel changes, and their causes, should be improved by further investigation.

**Keywords:** channel adjustments; incision; narrowing; human interventions; Iranian rivers

---

## 1. Introduction

Understanding of channel adjustment and controlling factors is very important for the interpretation of current channel conditions, prediction of future changes, and process-based river management [1,2]. Channel adjustments are driven both by natural (climate driven) and human factors. Human impact has produced remarkable channel changes, such as narrowing, incision, and changes to channel pattern [3–9]. The most common human interventions include channelization e.g., [10,11], dams e.g., [12–16], sediment mining [17,18], and land-use changes e.g., [19–35]. Natural factors include short-term climate changes e.g., [36], volcanic eruptions e.g., [37–39], large floods e.g., [40–42], and fires e.g., [10,43,44].

Changes in river channels can have social and environmental effects, it can damage structures, increase the potential for flooding, decrease groundwater resources, and cause a reduction in the number of native species. Therefore, a better understanding of channel adjustments is crucial for river management, and to predict and mitigate the negative effects of such adjustments.

During the last few decades, river channels in Iran have been stressed by climate change and several human interventions, such as channelization, the diversion of water for flood mitigation and agriculture, damming, gravel and sand mining, and land-use changes. Although studies have been

carried out on channel adjustments in Iranian rivers, a comprehensive review at national scale is lacking. By reviewing published articles and data, we addressed the following questions: (1) How much Iranian rivers have changed over the last decades (type and magnitude of change)? (2) What are the main causes of channel adjustments? (3) Are adjustments homogeneous throughout Iran or do some differences exist between the main geographic regions? Finally, channel adjustments of Iranian rivers were considered in a larger context, by comparing the situation in Iran with adjustments in other regions worldwide.

## 2. Study Area

The territory of Iran has an area of 1,648,000 km$^2$, and is located at 25° to 40° northern latitude and 44° to 64° eastern longitude (Figure 1). Except the narrow plains along the Caspian Sea, the Persian Gulf, and Khouzestan Region, Iran is considered as a highland with an average elevation of 1000 m a.s.l. The lowest point in Iran is located in the northeast, in the salt plain of Shahdad on the margin of Lut Kavir, with an elevation of 350 m. The highest point in Iran is Mount Damavand in the Alborz Mountain range, with a height of 5774 m. From a global tectonic point of view, Iran is part of the Alpine–Himalayan orogenic belt that extends from the Atlantic Ocean to the Western Pacific.

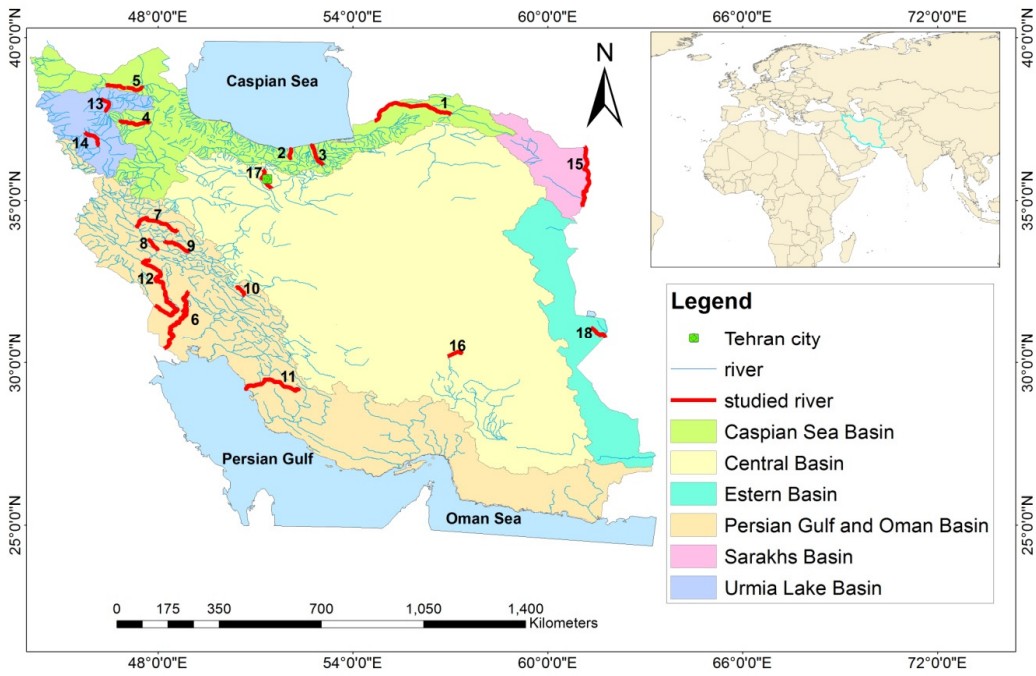

**Figure 1.** Location map showing Iranian river network, main drainage basins, and rivers analyzed in this study.

Iran features very diverse climatic and geographic characteristics. The highest temperature in the Persian Gulf area can reach up to 53 °C in summer, and the lowest temperature in the northwestern part of Iran can reach up to −40 °C in winter [45]. The average annual precipitation is estimated at 250 mm, varying from 50 mm in some areas of the central water basin, to more than 1600 mm in some coastal areas near the Caspian Sea [46]. The climate ranges from very humid to very arid, but in the majority of the country it is very arid and arid (Figure 2).

According to the classification of the Ministry of Energy, Iranian rivers can be grouped into six major drainage basins: 1—Caspian Sea Basin; 2—Persian Gulf and Oman Sea Basin; 3—Urmia Lake Basin; 4—Central Basin; 5—Eastern Basin (Hamoon); 6—Sarakhs Basin. All these basins are interior, except for the Persian Gulf and Oman Sea Basin. We found information about channel adjustments for 18 rivers, which are mainly located in the Persian Gulf and Oman Sea Basin (seven rivers) and the Caspian Basin (five rivers) (Figure 1). The main hydrological and physiographic characteristics of the

rivers analyzed in this study are summarized in Table 1. The hydrological regime of all studied rivers is perennial.

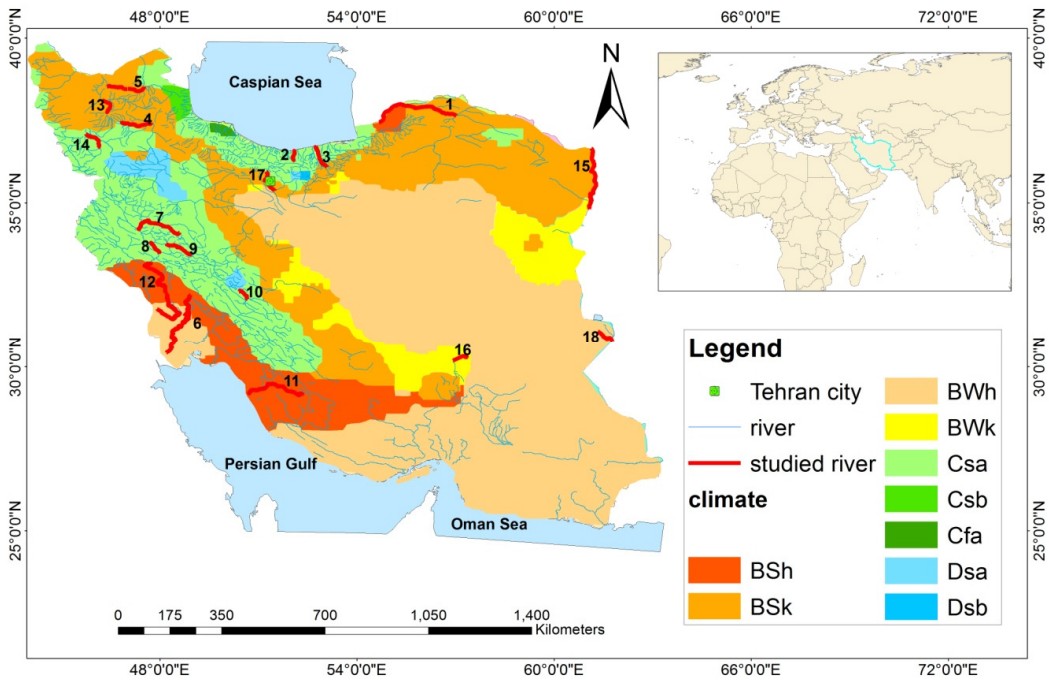

**Figure 2.** Climatic condition of Iran (1990 to 2014) according to the Koppen–Geiger classification (BSh: Semi-arid (steppe) desert and hot; BSk: Semi-arid (steppe) desert and cold; BWh: Arid desert and hot; BWk: Arid desert and cold; Csa: Mediterranean hot summer; Csb: Mediterranean warm/cool summer; Cfa: Humid subtropical; Dsa: Hot summer continental; Dsb: Warm summer continental or hemiboreal) [47].

**Table 1.** Hydrological and physiographic characteristics of the studied rivers (from: [48–65]).

| ID | River | Drainage Basin Area (km$^2$) | Length (km) | Basin Relief * (m) | Mean Annual Precipitation (mm yr$^{-1}$) | Mean Annual Discharge (m$^3$ s$^{-1}$) |
|---|---|---|---|---|---|---|
| 1 | Atrak | 27,546 | 214 | 2508 | 350 | 12.6 |
| 2 | Lavij | 146 | 38 | 3200 | 600 | 1.7 |
| 3 | Talar | 2478 | 121 | 3782 | 820 | 13.7 |
| 4 | Qaranqu | 3593 | 190 | 2637 | 374 | 8 |
| 5 | Ahar Chai | 3035 | 132 | 2864 | 330 | 1.3 |
| 6 | Karoon | 60,737 | 800 | 4536 | 660 | 400 |
| 7 | Gamasiyab | 7770 | 270 | 2219 | 420 | 150 |
| 8 | Kashkan | 9120 | 231 | 2500 | 550 | 33.2 |
| 9 | Horroud | 1123 | 83 | 2000 | 500 | 2.5 |
| 10 | Khoshkehroud | 250 | 17 | 1000 | 461 | 66 |
| 11 | Dalaki | 5210 | 150 | 3000 | 325 | 13.7 |
| 12 | Karkheh | 51,482 | 900 | 3642 | 477 | 167 |
| 13 | Lighvan Chai | 142 | 28 | 1690 | 292 | 0.8 |
| 14 | Zarrineroud | 11,729 | 217 | 2000 | 330 | 62.1 |
| 15 | Harriroud | 70,600 | 900 | 950 | 188 | 22.3 |
| 16 | Dehbala-Kerman | 89 | 20 | 1500 | 153 | - |
| 17 | Kan | 224 | 33 | 2496 | 300 | 2.7 |
| 18 | Sistan | 2500 | 72 | 14 | 52 | 69.4 |

* Basin relief is a difference between maximum elevation in the basin and elevation at basin outlet.

## 3. Materials and Methods

A review of studies about morphological changes in Iranian rivers was carried out. It is worth noting that this is the first review about this topic in Iran. We looked for studies, both in English

and Persian, dealing with channel changes during the last few decades. The review was focused on: (1) Type and magnitude of morphological changes, (2) causes of changes, (3) effects of changes on structures and environment.

Data were found for 18 rivers and allowed for a description of each river by the following aspects: Channel morphology; morphological changes; location and time of changes; causes of changes. Such aspects were defined after a preliminary screening of available data, taking into account possible limitations (e.g., lack of information) of the studies that we analyzed. As for channel morphology, we opted for just two typologies, single-thread and multi-thread. Morphological changes refer mainly to change in channel width (narrowing and widening) and in bed-level (incision and aggradation). The time of change (the period of time when channel changes have taken place) is crucial information since it allows for an assessment of the magnitude of processes that have occurred. Causes of changes are those mentioned in the studies, therefore we relied on original interpretations for this aspect, as well as for the others. For sediment mining, one of the major causes of changes in Iranian rivers, was if activity was carried out within the channel (in-stream mining). If mining activity was not clearly described, not indicating if activity was carried within the channel or in the fluvial corridor, this was simply reported in our analysis as 'mining'. Climate was defined for each river, as it was an aim of this study to investigate the possible role varying climate has, specifically if relation to channel adjustments (e.g., arid vs. temperate climate).

Finally, to facilitate analysis and interpretation of the dataset, a numerical identifier (ID) was associated to each river or river segment.

## 4. Results

### 4.1. Type and Magnitude of Channel Changes

As for planform changes, both narrowing and widening have occurred in cases 13 and 7, respectively (Table 2). Narrowing ranged between 19% and 84% (Figure 3) and, in some cases, was shown to be up to 300 m (e.g., some reaches of the Karun River). Widening ranged between 22% and 349% and has been shown to be more than 100 m in some rivers (e.g., Gamasiyab and Harirroud rivers). Furthermore, changes in channel pattern have occurred in some reaches (Table 2): From braided to meandering and then straight (e.g., Ahar Chai River); from meandering to braided (e.g., Gamasiyab River).

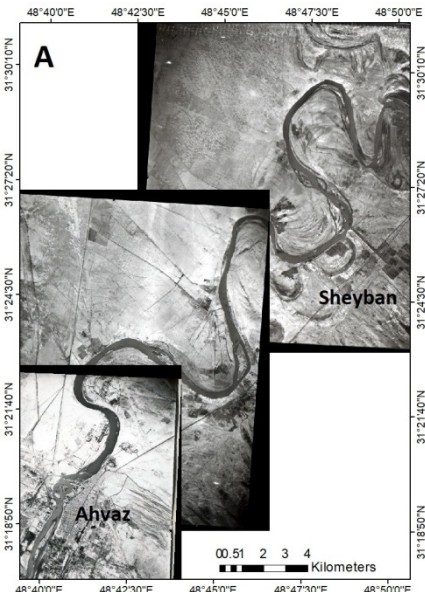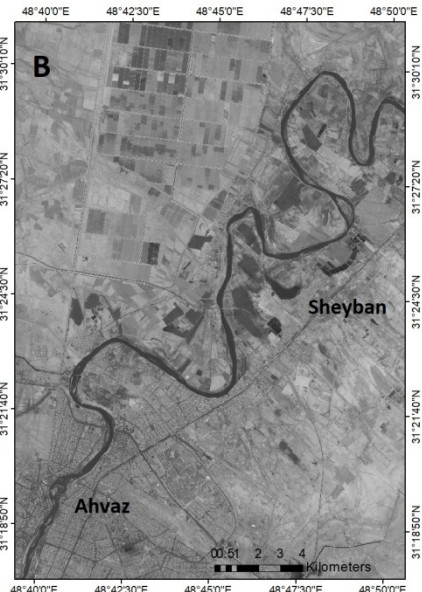

**Figure 3.** Channel narrowing in the Karun River due to human intervention, especially dam construction [66]: (**A**) aerial photograph of 1954; (**B**) IRS imagery of 2006.

As for bed-level changes, incision was observed in eight rivers, while aggradation was seen in three rivers (Table 2). Incision has been commonly seen at 1 to 2 m, but in some reaches, up to 6 to 7 m was seen (e.g., Talar and Khoshkehroud rivers) (Figure 4). An estimate of aggradation is not available.

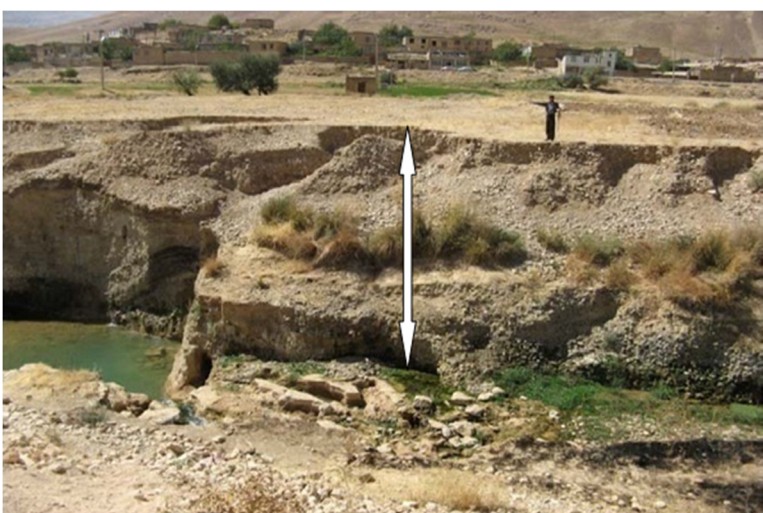

**Figure 4.** Channel incision in the Khoshkehroud River [65].

Coupling of processes, both planform and bed-level changes, was available for ten cases out of 21 (Table 2). Narrowing has been associated with incision and aggradation in cases 5 and 2, respectively, while widening with incision and bed-level stability was seen in case 2 and 1, respectively.

Some channel changes started in the 1950s to 1960s, whereas some others begin at the beginning of this century. This suggests that, overall, the morphological changes have been quite intense since they occurred in a time period of 50 to 60 years and, in some cases, of only 15 to 20 years.

**Table 2.** Channel adjustments in Iranian rivers and relative causes.

| ID | River | Channel Morphology | Morphological Changes | Location and Time of Changes | Causes | Climate Type (Koppen-Geiger Classification) | Reference |
|---|---|---|---|---|---|---|---|
| | | | | **Caspian Sea Basin** | | | |
| 1 | Atrak | Single thread | Widening (up to 11 m), depth (no changes) | Alluvial plain reaches; 1967 to 2001 | Changes of hydrology; changes of land use and cultivation pattern in floodplain | BSk | [62] |
| 2 | Lavij | Single thread | Widening (1 to 24 m); incision (0.2 to 1.8 m) | Alluvial plain reaches (150 km); 2008 to 2011 | In-stream gravel and sand mining; channelization; river engineering | Csa | [52] |
| 3a * | Talar | Single thread and multi-thread | Incision (3 m in most reaches) | Alluvial plain reaches (90 km); 1971 to 2005 | Gravel and sand mining | Csa | [57] |
| 3b * | Talar | Single thread and multi-thread | Narrowing (on average 84%), braided index decreased and the sinuosity index increased | 1955 to 2013 (11.5 km) | Land-use changes from forest land and riparian vegetation to residential lands | Csa | [67] |
| 3c * | Talar | Single thread and multi-thread | Narrowing (on average 25.5 m), aggradation | 1968 to 2013 | Land-use changes increases of 192% and 622% have been observed for orchards and residential areas; forest and riparian vegetation decreased | Csa | [68] |
| 4 | Qaranqu | Single thread | Widening (70 to 100 m in some cases); incision (0.75 to 2.5 m) | Mountain reach (20 km); 2008 to 2013 | In-stream gravel and sand mining; Dam | BSk | [49] |
| 5 | Ahar Chai | Single thread | Incision (on average 0.6 m downstream of dam); narrowing (on average 8 m downstream of dam); from braided to meandering and then straight | Mountain reaches (35 km); 1978 to 2005 | Dam | BSk | [50] |

**Table 2.** *Cont.*

| ID | River | Channel Morphology | Morphological Changes | Location and Time of Changes | Causes | Climate Type (Koppen-Geiger Classification) | Reference |
|---|---|---|---|---|---|---|---|
| | | | | **Persian Gulf and Oman Basin** | | | |
| 6a * | Karoon | Single thread | Narrowing (on average 143 m), aggradation | Mollasani to Farsiat; Alluvial plain reaches (110 km); 1995 to 2011 | Dam; bridge; land-use changes in floodplain; human interventions | BWh | [66] |
| 6b * | Karoon | Single thread | Narrowing (on average 18 m) | From Gotvand to Shoshtar; Alluvial plain reaches (128 km); 1989 to 2008 | Land-use and land-cover changes; dam; gravel mining | BSh | [34] |
| 7 | Gamasiyab | Single thread and multi-thread | Widening (on average 122 m); changes in channel pattern (from meandering to braided) | Alluvial plain reach; 1955 to 2010 | Fluctuation peak discharges; land-use changes | Csa | [58] |
| 8 | Kashkan | Single thread and multi-thread | Narrowing in most reaches (6 to 66 m) | Mountain reach (14 km); 2002 to 2009 | In-stream gravel and sand mining; bridge; diversions; changes of hydrology; hydraulic structures | Csa | [61] |
| 9 | Horroud | Single thread and multi-thread | Widening (on average 20 to 37 m in alluvial reaches and 1 m in mountain reach) | Mountain and alluvial plain reaches (83 km); 1955 to 2007 | Human interventions and development of cultivation and residential land | Csa | [63] |
| 10 | Khoshkehroud | Single thread | Incision (1 to 6.5 m); narrowing (5 to 40 m) | Alluvial plain (5 km); 2001 to 2006 | In-stream gravel and sand mining | Csa | [65] |
| 11 | Dalaki | Single thread | Narrowing (on average 23 m in most reaches) | Mountain and alluvial plain reaches; 1975 to 2013 | Natural factor (loess lithology and low slope, drought), dam; gravel and sand mining from floodplain | BSh | [54] |
| 12 | Karkheh | Single thread | Narrowing (on average 17 m); Incision on (average 0.1 m in most reaches) | Alluvial plain (218 km); 2002 to 2014 | Dam | BSh & BWh | [48] |

**Table 2.** *Cont.*

| ID | River | Channel Morphology | Morphological Changes | Location and Time of Changes | Causes | Climate Type (Koppen-Geiger Classification) | Reference |
|---|---|---|---|---|---|---|---|
| colspan | | | Urmia Lake Basin | | | | |
| 13 | Lighvan Chai | Single thread | Incision (up to 1 m); narrowing (up to 18 m) in few cross sections in the upstream part of the study reach, widening in few cross sections in the downstream part. | Mountain reaches (15 km); 2000 to 2012 | Channelization, increase of peak discharges (due to an increase of precipitation and land-use changes) | BSk | [55] |
| 14 | Zarineroud | Single thread | Narrowing (13 m) | Alluvial plain reach; 1955 to 2003 | Dam | Csa | [51] |
| colspan | | | Sarakhs Basin | | | | |
| 15 | Harirroud | Single thread and multi-thread | Widening (on average 115 m) | Mountain and alluvial plain reaches; 1974 to 2011 | Increasing discharge and sediment; land-use changes in floodplain; human interventions | BSk | [53] |
| colspan | | | Central Basin | | | | |
| 16 | Dehbala-kerman | Single thread | Widening (on average 84 m) | Mountain and alluvial plain reaches (10 km); 1988 to 2011 | In-stream gravel and sand mining | BWk | [56] |
| 17 | Kan | Single thread | Incision (up to 2 m); narrowing | Alluvial plain reaches; 1969 to 2009 | In-stream gravel and sand mining | BSk | [59] |
| colspan | | | East Basin | | | | |
| 18 | Sistan | Single thread | Narrowing (on average 183 m) | Alluvial plain reaches; 1956 to 2008 | Dam | BWh | [64] |

* a, b, and c are the different reaches of the same river.

### 4.2. Causes and Effects of Channel Adjustments

All the analyzed studies made an attempt to identify the causes of channel adjustments. Both anthropic (e.g., sediment mining, dams, land-use changes) and natural (climate change) causes have been recognized. Channel adjustments have been related to a combination of different causes in 11 cases, whereas to a single cause in ten cases (Table 2). As for multiple causes, a combination of anthropic and natural factors were identified in five cases, whereas a combination of different anthropic factors were identified in six cases. As for single causes, both sediment mining and dams have been identified in four cases, whereas land-use change was identified in two cases.

The next step in our analysis of the dataset was to identify the main causes of adjustments. Being aware of some limitations, namely that we had to select a main factor for those rivers characterized by multiple causes, and the relatively small size of the dataset, this analysis could be useful to explore possible relationships between causes (e.g., controlling factors) and channel adjustments. Three causes turned out to be the dominant causes in 16 out of 18 rivers (Figures 5 and 6): Gravel and sand mining (river ID: 2, 3, 4, 8, 10, 16, 17); dams (ID: 5, 6, 12, 14, 18); climate change (ID: 7, 9, 11, 15). Channelization and land-use change (coupled with climate change) were identified for the other two rivers of the dataset. If channel response to gravel and sand mining is straightforward in terms of bed-level change (all the available case studies show mining associated with channel incision; Figure 6), it is not in terms of planform change. In fact, the response to sediment mining was both narrowing and widening (Figure 5). On the contrary, the response to dams seems more direct, since narrowing and incision have been observed in all the rivers with such human disturbance. Finally, climate change seems to mainly induce channel widening (Figure 5).

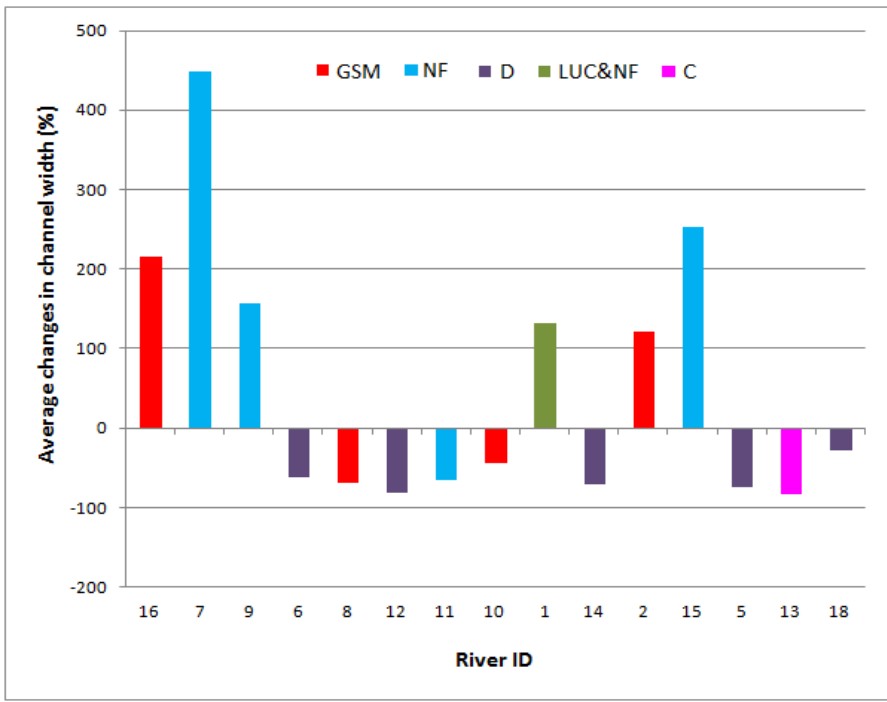

**Figure 5.** Changes in channel width and relative causes in Iranian rivers. GSM: Gravel and sand mining, NF: Natural factors, D: Dam, LUC&NF: Land-use change and natural factors, C: Channelization.

Morphological changes in river channels may have negative effects on hydraulic structures, infrastructure, agricultural lands, and the environment. Gravel and sand mining has led to the instability of banks, damage to structures, increasing flood hazard and potential damages to bridges (Figure 7). Widening of the channel has led to a loss of agricultural land (e.g., Harirroud River). Finally, it is likely that channel changes have had effects on environment and ecosystems, but such effects were not evaluated in the available studies.

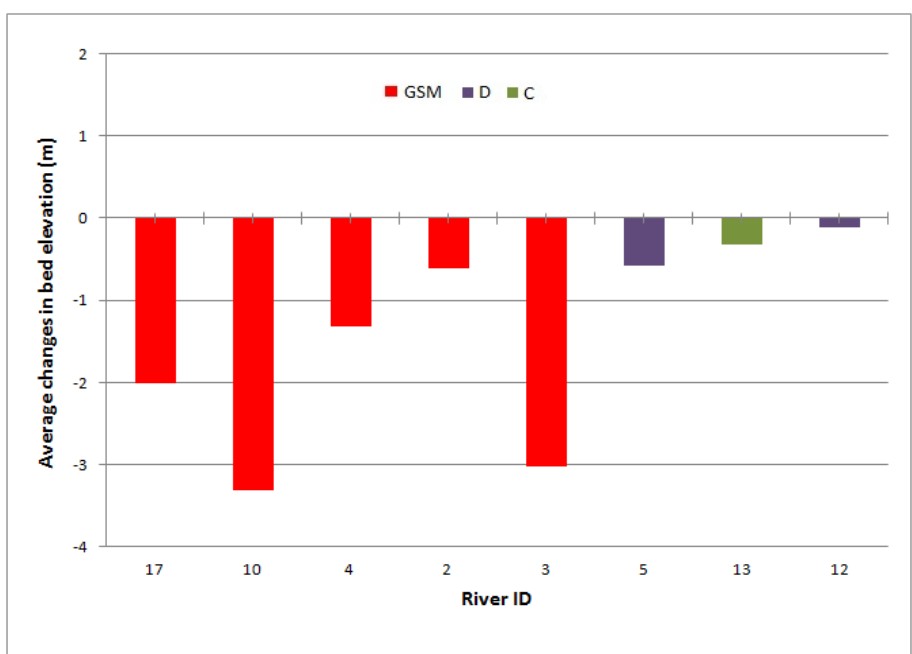

**Figure 6.** Changes in bed elevation and relative causes in Iranian rivers. GSM: Gravel and sand mining, D: Dam, C: Channelization.

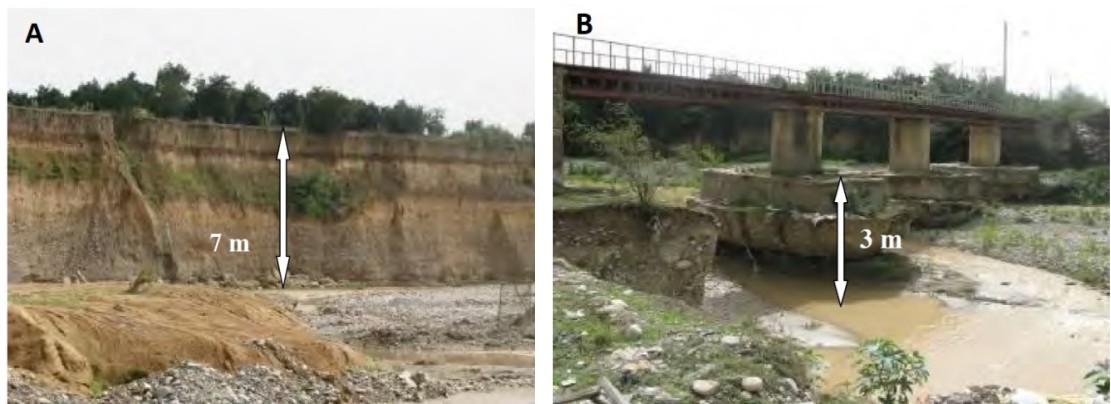

**Figure 7.** Channel incision due to in-stream gravel and sand mining (**A**); Effects of channel incision along the Talar River and effects on bridge stability (**B**) [57].

## 5. Discussion

### 5.1. Channel Adjustment in Iranian Rivers and Comparison with Other Rivers Worldwide

Channel adjustment has been remarkable in Iranian rivers (a narrowing up to 73% and incision up 6 to 7 m). These channel adjustments bring to mind the question whether such adjustments in Iran are similar to those that have occurred in other rivers worldwide. Studies about rivers in Europe (e.g., Italy, France, Spain, UK, Poland, and Hungary), and elsewhere (China and USA) show that narrowing and incision have been the dominant processes in these rivers, and that sediment mining, damming, channelization, over grazing, and land-use changes are the main controlling factors (Table 3). As for planform adjustments, narrowing was the dominant process in Iranian rivers, but it is worth noting that widening took place in 7 out of 20 cases (Table 2 and Figure 3). Similarly, in regard to bed-level changes, incision was dominant, but aggradation occurred in 3 out of 11 cases (Table 2 and Figure 4). As for controlling factors, sediment mining and damming are the most important in Iranian rivers, as

well as in several rivers worldwide, although climate change turned out to be the key factor in 4 out of 18 rivers (Figures 5 and 6).

**Table 3.** Channel changes due to human activities: Case studies from different countries and environments.

| Region | Channel Adjustments | Causes | Climate Type (Koppen-Geiger Classification) | Reference |
|---|---|---|---|---|
| Spain | Incision (more than 2 m) narrowing, changes in channel pattern (from braided to meandering) | Reforestation and expansion of shrubs, depopulation and farmland abandonment resulting in plan recolonization in formerly cultivated areas | Cfb * | [69] |
| Spain | Incision (4 to 6 m); changes in channel pattern (from braided to straight); in some reaches widening and some reached narrowing | Gravel mining; dams; afforestation; water withdrawal for irrigation | Cfb | [18] |
| Europe | Narrowing; incision; changes in channel pattern (from braiding to wandering or single thread) | Human interventions: Gravel and sand mining | Cfb | [3] |
| China | Incision (1.6 to 11 m, >10 m in the deepest cut); narrowing (up to 150 m at Sanshui) | Declining rainfall; dam constructions; water diversion; reforestation and afforestation; sediment mining | Cfb | [70] |
| New Mexico | Narrowing (5 to 103 m) | Decreasing peak flows; the implementation of flood control and engineering works; climate change, droughts | BSk | [71] |
| Scotland | Narrowing (on average 34%); incision | Flood embankment construction | Cfb | [72] |
| Northern England | Incision (average 1 m and up to 2 m); narrowing | Gravel extraction, magnitude flood with high frequency | Cfb | [73] |
| Poland | Incision (1 to 3.8 m); narrowing (10% up to 70%) | Channelization; decrease in sediment supply; in-stream gravel mining; dam, land-use changes | Dfb ** | [74,75] |
| Hungary | Incision (up to 3.8 m); narrowing (up to 50 m); width of the river decreased by 17% to 45%, while its mean and maximum depth increased by 5% to 48% | Channel cut-offs; revetments and groynes | Csc ***/Cfa | [76] |
| USA | Widening (up to 50%); aggradation (less than 1 m) | Hard-rock mining; road construction, timber harvest, heavy grazing of uplands, land-use change | Csb/Dsb | [77] |
| France | Incision (up to 5 m); narrowing (up to 22 m) | Heavy grazing; land-use change; gravel mining; check dams | Cfb | [31,77] |
| Italy | Incision (3 to 4 m and in some cases more than 10 m); narrowing (up to 50% or more); changes in channel pattern (from braided to wandering) | Gravel and sand mining; dams; channelization | Cfa/Cfb/Csb/Csa/Csc/BSk | [11] |

\* Oceanic climate; \*\* Warm-summer humid continental climate; \*\*\* Mediterranean cold summer climates.

This comparison suggests that Iranian rivers share several aspects with other rivers worldwide (e.g., type of adjustments, controlling factors). At the same time, some aspects are worthy of further detailed exploration, specifically the role of climate change on Iranian rivers.

Observations unequivocally show that Iran has been rapidly warming over recent decades, triggering a wide range of climatic changes. Iran has warmed by nearly 1.3 °C during the period 1951 to 2013 (+0.2 °C per decade), with an increase of the minimum temperature at a rate two times that of the maximum. Consequently, an increase in the frequency of heat extremes and a decrease in the frequency of cold extremes have been observed. The annual precipitation has decreased by 8 mm per decade, causing an expansion of Iran's dry zones. Previous studies have pointed out that warming is generally associated with more frequent heavy precipitation because warmer air can hold

more moisture [78]. The percentages of areas with extremely dry and dry climate have increased by nearly 8% and 6%, respectively, while a decreased percentage of Iran is now characterized by relatively wet climate [78,79]. Dry zones have been pushed into the southwest and northeast of Iran, which previously had a mostly semi-dry climate. Some areas in south Iran with a dry climate in the past have recently been characterized by an extremely dry climate. West Iran, which was previously characterized by a semi-wet climate, has recently become mostly semi dry [78]. It seems that the Central Basin and Persian Gulf and Oman Sea Basin and Sarakhs Basin have been affected by climate change, with as dryer climate being more common than other basins. That is why climate change is likely the major controlling factor of channel adjustments in these basins (e.g., rivers 7, 9, 11, and 15).

*5.2. Dataset Limitations and Research Perspectives*

It is worth noting that this work represents the first review about channel adjustments in Iranian rivers, and data turned out to be neither homogeneous nor complete for all the rivers. Some major limitations of the available dataset are: (i) Few data about bed-level changes, especially about channel aggradation, (ii) lack of reconstruction of evolutionary trajectory, and (iii) few case studies in some basins (e.g., Central, Eastern and Sarakhs basins). This implies that understanding channel adjustments, and their causes, could be improved notably by further studies. Reconstruction of evolutionary trajectory will be crucial for a sound understanding of factors controlling adjustments and for river management [80]. Moreover, the role of climate change in the recent evolution of Iranian rivers is a very challenging issue that future studies should address.

## 6. Conclusions

(1) Remarkable channel adjustments took place in Iranian rivers during the last few decades. Adjustments seem comparable with other rivers worldwide, although lack of evolutionary trajectories hinders a detailed analysis of timing and magnitude of morphological changes.

(2) Human interventions, specifically sediment mining and dams, have been the main causes of channel adjustment in several rivers, though climate change has played a major role in some rivers.

(3) Besides analyzing type, magnitude, and causes of channel adjustments, this review turned out to be useful for the identification of key issues that should be addressed in the future. More detailed studies will allow a better understanding of controlling factors, specifically to what extent climate change is driving the evolutionary trajectory of Iranian rivers.

**Author Contributions:** S.K. and N.S. designed the research. S.K. collected the data, made the first analyses, and wrote the first draft of the manuscript; N.S. contributed interpretation of the dataset and substantive manuscript writing.

**Funding:** This research was supported by Shahid Beheshti University G.C. with contract number S/600/688.

**Conflicts of Interest:** The authors declare no conflict of interest.

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
