# Peer review of "Channel Adjustments in Iranian Rivers: A Review"

_water, doi:10.3390/w11040672_

Round 1
Reviewer 1 Report
Dear authors,
you present an interesting manuscript from underevaluated environments and I think that you review is worth to publish. I see rather some minor shortcomings, which could increase the quality of your manuscript:
General comments:
- I do not understand differences between the 'in-channel minning' and 'sediment (or gravel or sand) mining' (e.g. Table 2 and elsewhere in the text) - please explain in the introduction or methods. Otherwise I have no idea how can sediment mining outside fluvial corridor affect channel processes.
- it is possible to propose some general scheme (or a few general schemes for particular regions) which will graphically illustrate recent trends in Iranian rivers (e.g., their evolution trajectory)? I think that such scheme would be very demonstrative to express a lot of information mentioned in Table 2 and the paper could get more potential to be highly referred in the future. For some inspiration, see e.g. Liébault et al, 2005, River Research and Applications (10.1002/rra.880), Figure 1.
- general conclusion is missing at the end of your manuscript. Please spend a few sentences to describe main observations and to explain how Iranian rivers differ from global trends.
Specific comments:
l. 15 - I would put exact types of natural factors (e.g. change in precipitation etc.)
l. 69-74 - I would delete this part - I think that it is really not necessary to mention all important rivers in Iran
l. 150-155 - this part should go to discussion
Table 1 - what does 'basin relief' means? Mean or maximal elevation in the basin? Please add missing numbers (I guess that missing elevations can be easily obtained from topographic maps)
- I would also add sources of information in this table (e.g. for mean precipitations and discharges, I do not think that you measured or calculated them)
- I would add brief information about hydrological regime of the studied rivers - please categorize them as perennial, intermittent and ephemeral based on the flow occurrence
Table 2 - ID1 - please remove 'changes in grain size...' - this is a consequence of causes, not a cause
Figure 1 - please arrange the rivers by causes or by magnitude of observed changes
Best wishes
Author Response
Reviewer 1
Dear authors,
You present an interesting manuscript from underevaluated environments and I think that you review is worth to publish. I see rather some minor shortcomings, which could increase the quality of your manuscript:
General comments:
- I do not understand differences between the 'in-channel minning' and 'sediment (or gravel or sand) mining' (e.g. Table 2 and elsewhere in the text) - please explain in the introduction or methods. Otherwise I have no idea how can sediment mining outside fluvial corridor affect channel processes.
Since only some works describe clearly how sediment mining was carried out, we preferred to indicate if mining was surely carried out within the channel (i.e. in-channel mining). We added a sentence to clarify this point in the “Materials and Methods” section.
- it is possible to propose some general scheme (or a few general schemes for particular regions) which will graphically illustrate recent trends in Iranian rivers (e.g., their evolution trajectory)? I think that such scheme would be very demonstrative to express a lot of information mentioned in Table 2 and the paper could get more potential to be highly referred in the future. For some inspiration, see e.g. Liébault et al, 2005, River Research and Applications (10.1002/rra.880), Figure 1.
Thanks for this very useful suggest. Unfortunately, we think that it is not possible to propose a general scheme since channel adjustments are not homogeneous in Iranian rivers (i.e. narrowing and incision were dominant, but also widening and aggradation took place) and only with a larger dataset it could be possible to propose one or more general schemes.
- general conclusion is missing at the end of your manuscript. Please spend a few sentences to describe main observations and to explain how Iranian rivers differ from global trends.
Thanks for this suggest. We added a new section (“Conclusions”) to summarize the main outcomes of this work.
Specific comments:
l. 15 - I would put exact types of natural factors (e.g. change in precipitation etc.)
Done
l. 69-74 - I would delete this part - I think that it is really not necessary to mention all important rivers in Iran
Done
l. 150-155 - this part should go to discussion
We prefer to keep this part in the “Results” section, since it refers still to analysis of the data and, specifically, to effects of channel adjustments on structures, environments, etc.
Table 1 - what does 'basin relief' means? Mean or maximal elevation in the basin? Please add missing numbers (I guess that missing elevations can be easily obtained from topographic maps)..
Basin relief is the a difference between maximum elevation in the basin and elevation at basin outlet.
We added
- I would also add sources of information in this table (e.g. for mean precipitations and discharges, I do not think that you measured or calculated them)
Done
- I would add brief information about hydrological regime of the studied rivers - please categorize them as perennial, intermittent and ephemeral based on the flow occurrence
Done
Table 2 - ID1 - please remove 'changes in grain size...' - this is a consequence of causes, not a cause
Done
Figure 1 - please arrange the rivers by causes or by magnitude of observed changes
Thanks for this suggest. We thought about different ways of representing the analysed rivers, but we decided that it is not feasible to represent rivers according to causes (although we try to identify a main cause for each river this is a simplification of real conditions) or magnitude. Besides, this figure represents a general setting, data are analysed in the following sections of the paper.

Reviewer 2 Report
The article is good. The authors present their considerations clear and understandable. Based on available data the conclusions are correct. It is very important that the authors critically refer to the results, having in their mind the real low data availability.
I thing, it is worth to published this article as a review as it was indicated by authors. I have only one small remark: could you please put the names or ID's of rivers on to fig 1, it will improve the reading of the article.
Author Response
Reviewer 2
The article is good. The authors present their considerations clear and understandable. Based on available data the conclusions are correct. It is very important that the authors critically refer to the results, having in their mind the real low data availability.
I thing, it is worth to published this article as a review as it was indicated by authors. I have only one small remark: could you please put the names or ID's of rivers on to fig 1, it will improve the reading of the article.
Done
